# Visible light mediated photocatalytic [2 + 2] cycloaddition/ring-opening rearomatization cascade of electron-deficient azaarenes and vinylarenes

Noelia Salaverri[1], Rubén Mas-Ballesté [2,3], Leyre Marzo [1✉] & José Alemán [1,3✉]

The broad presence of azaarene moieties in natural products has promoted the development of new functionalization reactions, giving access to larger libraries of bioactive compounds. The light promoted [2 + 2] photocycloaddition reaction to generate cyclobutanes has been extensively studied in photochemistry. In particular, De Mayo reported the [2 + 2] cycloaddition followed by retroaldol condensation between enols of 1,3-dicarbonyls and double bonds to synthesize 1,5-dicarbonyls. Herein, we describe the [2 + 2] photo-cycloaddition followed by a ring-opening rearomatization reaction between electron-deficient 2-methylene-azaarenes and double bonds, taking advantage of the ability of these hetero-cyclic derivatives to form the corresponding *pseudo*-enamine intermediate. The procedure shows a high functional group tolerance either on the double bond or the heteroarene side and allows the presence of different electron-withdrawing groups. In addition, the wide applicability of this reaction has been demonstrated through the late-stage derivatization of several natural products. Photochemical studies, together with theoretical calculations, support a mechanism involving the photosensitization of the *pseudo*-enamine intermediate.

[1] Organic Chemistry Department, Módulo 2, Universidad Autónoma de Madrid, 28049 Madrid, Spain. [2] Inorganic Chemistry Department, Módulo 7, Universidad Autónoma de Madrid, 28049 Madrid, Spain. [3] Institute for Advanced Research in Chemical Sciences (IAdChem), Universidad Autónoma de Madrid, Madrid, Spain. ✉email: leyre.marzo@uam.es; jose.aleman@uam.es

Azaarene moieties such as the quinoxaline, pyrazine, quinoline or isoquinoline structures are a ubiquitous theme in natural products with interesting biological properties (Fig. 1)[1]. These scaffolds are present in a large number of pharmaceutical compounds, such as Timapiprant (an anti-asthmatic and anti-inflammatory agent)[2], Papaverine (a smooth muscle relaxant)[3,4], Emetine[5,6] (an anti-protozoan) and the quinoline derivative **I** (an atherosclerosis treatment)[7]. Other azaarenes such as quinoxaline derivatives **II** with anti-inflammatory properties, and pyrazines such as Terezine A with antifungal characteristics, are also prevalent biological compounds[8,9] (Fig. 1). In view of the wide application of electron-deficient azaarenes in drug discovery, the development of new methods to functionalize these structures are of considerable interest, affording new procedures to introduce such core in the skeleton of bioactive molecules and opening the door to the development of new libraries of bioactive compounds.

The [2 + 2] photocycloaddition of double bonds to generate cyclobutanes is one of the most employed reactions in photochemical synthesis[10–15]. The inherent strain of cyclobutanes makes them prone to undergo ring-opening reactions. In this context, de Mayo reported that, under UV irradiation, the enol of 1,3-dicarbonyl compounds involved in a [2 + 2] photocycloaddition with olefins to form a non-isolable cyclobutanol intermediate, which evolved to afford 1,5-diketones (X = O, EWG = COR) after retro-aldol condensation (Fig. 2a)[16,17]. The synthetic potential of this reaction was fully demonstrated in the rapid formation of complex molecular structures and natural product synthesis[18–24]. More recently, a visible light-mediated photocatalytic version of the reaction has been reported via photosensitization of styrene using the [Ir{dF(CF₃)₂ppy}₂(bpy)]PF₆ complex[25]. This approach solved the selectivity problem associated with undesired secondary reactions linked to the use of UV-light as the irradiation source. Regarding the scope of both the classical de Mayo photochemical reaction[26] and the visible-light photocatalytic approach, the common restraint is that the first [2 + 2] cycloaddition step only takes place with enols or preformed enamines, thus limiting the scope of substrates that can participate in this reactivity.

We envisioned that other *pseudo*-enamine species with a heteroarene core in the structure could undergo a [2 + 2] cycloaddition in the presence of an olefin, thus giving access to the derivatization of compounds with pharmacological or biological interest (see imine-enamine equilibrium, Fig. 2b). Therefore, in this work we present the [2 + 2] cycloaddition reaction between β-electron withdrawing substituted azaarene derivatives and different double bonds followed by a ring-opening rearomatization

of the cyclobutane intermediate. DFT calculations together with fluorescence quenching studies provide evidences of which species play a key role in the mechanistic proposal.

## Results and discussion

**Optimization of the model reaction.** Taking into account that quinolines are electron-deficient heteroarenes and can form *pseudo*-enamine species, we chose **1a** as model substrate for the [2 + 2] cycloaddition reaction. Therefore we studied the reaction between the quinoline derivative **1a** and styrene **2a** in the presence of the iridium complex **3a** ($E_T$ = 62 kcal mol$^{-1}$)[27] as photocatalyst in CH₃CN and irradiating the mixture with blue LED (455 nm) for 17 h (entry 1, Table 1). To our delight, the product **4a** was obtained with a 38% yield, thus corroborating the initial hypothesis and opening the door to the development of this transformation with electron-deficient azaarenes. As expected, the presence of DBU as the base enhanced the reactivity increasing the formation of **4a** to an 83% yield (entry 2, Table 1). Other photocatalysts such us Ru(bpy)₃Cl₂ (**3b**, $E_T$ = 46 kcal mol$^{-1}$)[28] or the Fukuzumi´s catalyst (**3e**, $E_T$ = 44 kcal mol$^{-1}$)[29] with lower triplet energies did not show any reactivity, while Ir(ppy)₃ (**3c**, $E_T$ = 55 kcal mol$^{-1}$)[30] or 4CzIPN (**3d**, $E_T$ = 60 kcal mol$^{-1}$)[31] afforded the final product although in a lower yield than **3a** (entries 3–6, Table 1). Carrying out the reaction in THF, **4a** was isolated with a 98% yield (entry 7, Table 1). It should be highlighted that the reaction proceeded in a regioselective manner since the other plausible product coming from the enolization of the ester [CH₂-CO₂Me → CH = C(OH)OMe] was not observed (See Supplementary Note 5). In addition, the photocatalytic nature of the reaction was confirmed when the experiment was performed without light or a photocatalyst, without conversion for both cases (entries 8 and 9, Table 1).

**Substrate scope.** With the optimized conditions established (entry 7, Table 1), a variety of heteroarenes, different EWGs and double bonds were examined in the reaction (Fig. 3). The presence of halogens in different positions of the quinoline ring did not affect the reactivity (**4b**-**4d**), and excellent yields were obtained in all cases. The methodology also allowed the presence of electron-withdrawing groups (CO₂H, **1e**) or electron-donating substituents at the aryl moiety (MeO, **1f**). Next, other electron-withdrawing substituents in the β-position to the azaarene were studied (EWGs row, Fig. 3), obtaining good to excellent results for the quinoline derivatives with amide (**4g**), nitrile (**4h**), sulfone (**4i**) and arylketone (**4j**). Moreover, even in the case of the more

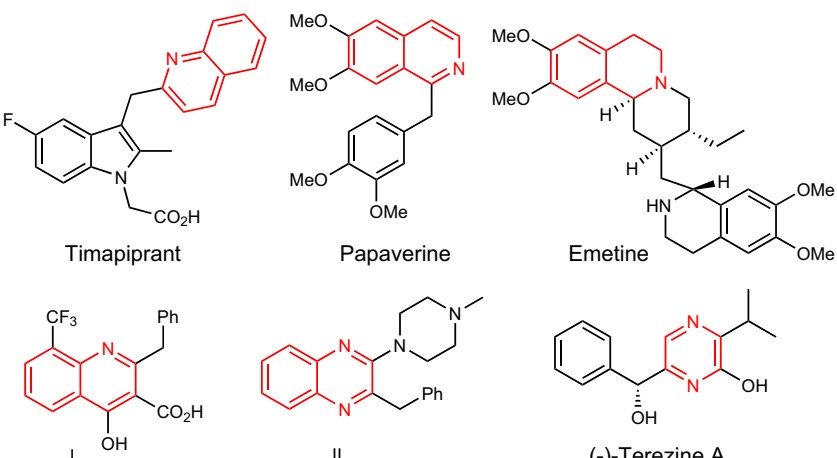

**Fig. 1 Biologically active compounds.** Compounds bearing different azaarene moieties with interesting properties.

enolizable arylketone **4j**, we observed exclusively one regioisomer. Furthermore, the reaction is not exclusive for 2-substituted quinolines but 4-substituted quinolines, isoquinolines, quinoxalines, and pyrazines also underwent this reactivity, affording the final products **4k-4o** in moderate to good yields (azaarenes row).

Later, the reaction between the quinoline **1a** and a variety of styrene derivatives **2** was studied (olefin row, Fig. 3). The presence of electron-donating substituents (**4p-4r**) and electron-withdrawing ones (**4s**) did not have a large influence on the final results, whereas the use of halogens (**4t**, **4u**) and *ortho* substituents (**4u** and **4v**) were also compatible. Pleasantly, the reaction also allowed the synthesis

of quaternary centres, starting from the α-methyl styrene and giving the **4w** quinoline derivative with a 70% yield. In addition, when β-methyl styrene derivatives were used, two new chiral centres were obtained with a complete diastereoselectivity for both cases (*anti*-**4x**, and *anti*-**4y**). The 2,3-dimethylbuta-1,3-diene previously employed in photocatalytic [2 + 2] cycloadditions[32], afforded the synthesis of the dialkylic quaternary centre product **4z** with a 70% yield. The reaction also took place with double bonds bearing electron-poor and electron-rich heteroarenes. Therefore, the reaction worked with the pyridine derivative (**4aa**) and the thiophene (**4ab**), with moderate to good yields. Interestingly, the indol derivative **4ac**, which is present in the Timapiprant core (an oral CRTH2 antagonist which has been proved to be an anti-asthmatic)[33] was isolated with a good yield. Some limitations were also observed such as the use of cyclohexene, free acidic protons at the aryl moiety in the double bond, or the use of stilbene (see limitations row).

In the field of drug discovery, late-stage functionalization (LSF) has become an excellent tool for the development of libraries of bioactive compounds, starting from lead structures and avoiding the de novo synthesis[34]. LSF facilitates the development of structure-activity relationships, the optimization of on-target potency and the improvement of the physical properties of the lead compounds. This is evidenced by the large number of methods for the functionalization of peptides[35–40] or the number of methodologies that have been recently developed for the functionalization of complex bioactive molecules[41–46].

Therefore, to prove the utility of our methodology, we introduced electron-deficient azaarenes in the structure of complex natural products in one step (Fig. 4). Thus, the hormone estrone was derivatized to **5a** with an 85% yield, as well as the amino acid Tyrosine derivative, that afforded **5b** with an excellent 89% yield. Finally, δ-Tocopherol (vitamin E), which is employed as a food preservative for its antioxidant properties, was transformed to **5c** with an excellent 92% yield.

**Mechanistic proposal**. Elucidation of the reaction mechanism was further addressed (Fig. 5). Based on a previous report[25], after photoexcitation of the photocatalyst **3** to its excited state **3***, energy transfer from **3*** to **2a** or the enamine **1a′** (that is almost

**Fig. 2 Precedents and this work. a** De Mayo reaction, **b** This work: [2 + 2] cycloaddition & ring-opening rearomatization.

**Table 1 Optimization of reaction conditions between the quinoline derivative 1a and styrene 2a.**

| Entry[a] | Photocatalyst (mol%) | Base | Solvent | Conversion [%][b] |
|---|---|---|---|---|
| 1 | [Ir{dFCF₃ppy}₂(bpy)]PF₆ (**3a**, 2) | – | CH₃CN | 38 |
| 2 | [Ir{dFCF₃ppy}₂(bpy)]PF₆ (**3a**, 2) | DBU | CH₃CN | 83 |
| 3 | [Ru(bpy)₃]Cl₂·6 H₂O (**3b**, 3) | DBU | CH₃CN | n.r. |
| 4 | [Ir(ppy)₃] (**3c**, 2) | DBU | CH₃CN | 80 |
| 5 | 4CzIPN (**3d**, 4) | DBU | CH₃CN | 43 |
| 6 | [Mes-Acr]ClO₄ (**3e**, 5) | DBU | CH₃CN | n.r. |
| 7 | [Ir{dFCF₃ppy}₂(bpy)]PF₆ (**3a**, 2) | DBU | THF | 100 (98)[c] |
| 8[d] | [Ir{dFCF₃ppy}₂(bpy)]PF₆ (**3a**, 2) | DBU | THF | n.r. |
| 9 | – | DBU | THF | n.r. |

n.r. no reaction.
[a]All the reactions were carried our using 0.1 mmol of **1a**, 0.5 mmol of **2a**, 0.5 equivalents of base and 1 mL of solvent, under 455 nm (22 W m⁻²) LED irradiation for 17 h, unless indicated otherwise.
[b]Determined by ¹H NMR using 1,3,5-trimetoxibencene as internal standard.
[c]Isolated yield.
[d]Without light. 4CzIPN = 1,2,3,5-tetrakis(carbazol-9-yl)-4,6-dicyanobencene.

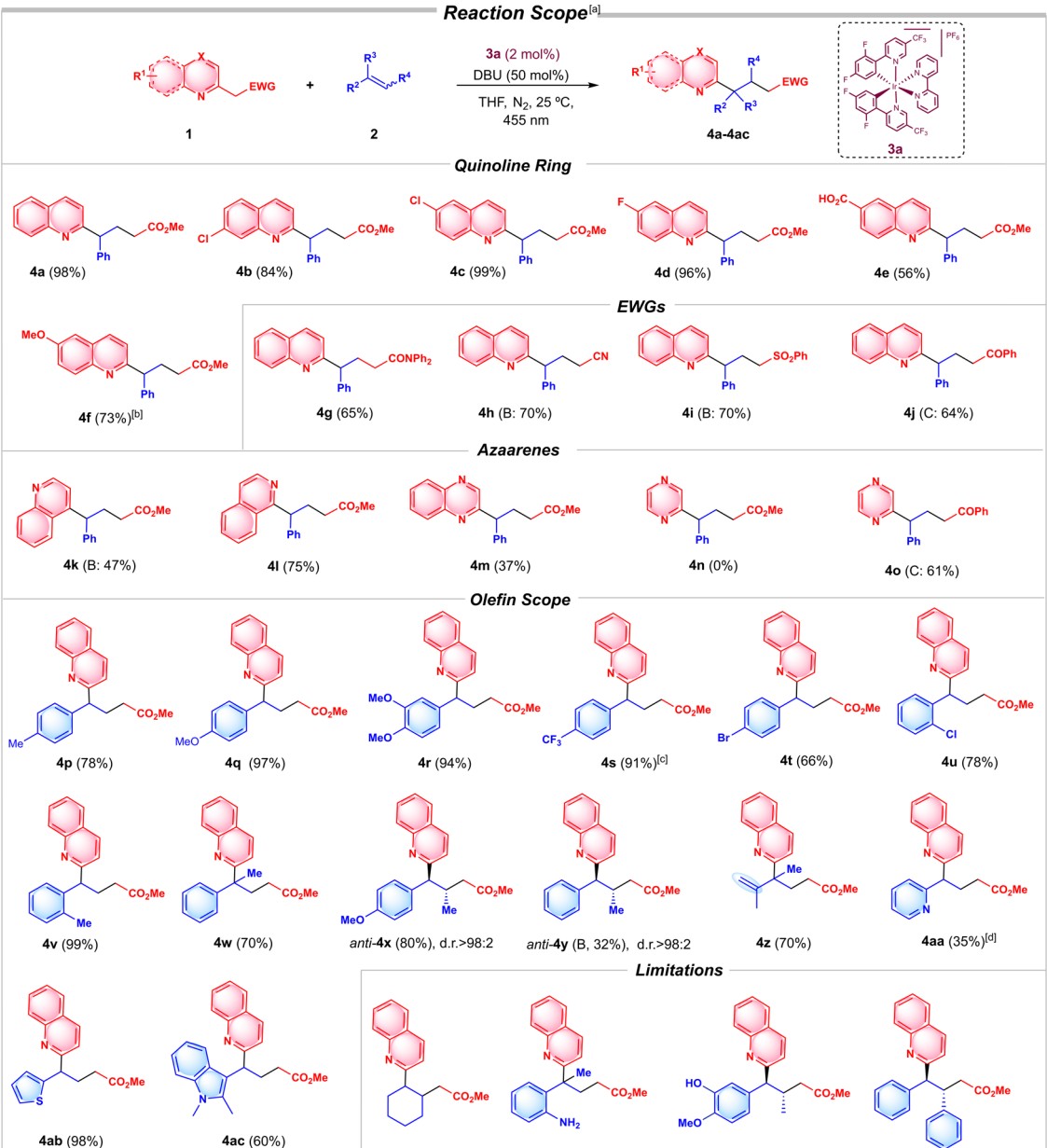

**Fig. 3 Scope of activated azaarenes 1 and olefins 2.** [a]Reaction conditions: All the reactions were carried out with 0.1 mmol **1**, 0.5 mmol **2**, under $N_2$, 25 °C, 455 nm LED irradiation (22 W m$^{-2}$), over 17 h and the following conditions: A: 0.002 mmol Ir[dFCF$_3$ppy]$_2$(bpy)PF$_6$ **3a**, 0.05 mmol DBU, THF (1 mL), except in the cases indicated with conditions B: 0.002 mmol Ir(ppy)$_3$ **3c**, 0.05 mmol DBU, THF (1 mL) or conditions C: 0.002 mmol Ir(ppy)$_3$ **3c**, 0.05 mmol K$_2$CO$_3$, CH$_3$CN (1 mL); [b]Reaction carried out over 30 h. [c]**4 s** was isolated as an inseparable mixture 5.2:1 of the adduct and the Michael addition product. [d]Lower yield is obtained due to the formation of the competing Michael adduct.

isoenergetic to the quinoline **1a**, see Supplementary Data 1) takes place to afford either $^3$(**1a'**)* or $^3$(**2a**)* (Fig. 5a). From the triplet excited state both $^3$(**1a'**)* or $^3$(**2a**)* undergo a [2 + 2] cycloaddition with **2a** or **1a'** respectively, yielding a common intermediate **I**. With regard to the experimental triplet energies described in the literature ($E_{T\ 2a}$ = 60.8 kcal mol$^{-1}$ [47]; $E_{T\ 3a}$ = 62.0 kcal mol$^{-1}$ [27]; $E_{T\ 3c}$ = 57.6 kcal mol$^{-1}$ [30]), styrene **2a** can be photosensitized by **3a**, but not by **3c**, whose triplet energy is lower than the triplet energy of **2a**. However, considering that the reaction with photocatalyst **3c** afforded **4a** with an 80% yield (entry 3, Table 1), photosensitization of **1a'** seems to be a probable mechanistic pathway.

Fluorescence quenching studies of **3a** and **3c** are consistent with the previous data, finding an efficient interaction of both **1a'**

and **2a** with **3a***, while in the case of photocatalyst **3c**, only the quinoline **1a'** was able to quench the excited state (see Fig. 5b and Supplementary Note 5). Therefore, our results indicate that the most probable mechanistic pathway is the photosensitization of **1a'**.

Further mechanistic insights were obtained by performing DFT calculations at the M062X/6–311 G** level of theory including solvation effects using the SMD model (solvent CH$_3$CN ε = 37.5) (Fig. 6 and Supplementary Table 1). According to the experimental results, the $S_0 \rightarrow T_1$ gap, considering the free energy of optimized geometries in both singlet and triplet spin states, is significantly lower for **1a** than for **2a** ($E_{T\ calculated\ 1a'}$ = 48.8 kcal mol$^{-1}$; $E_{T\ calculated\ 2a}$ = 54.9 kcal mol$^{-1}$) (Comparison between experimental and theoretical data for **2a** indicates that our calculations

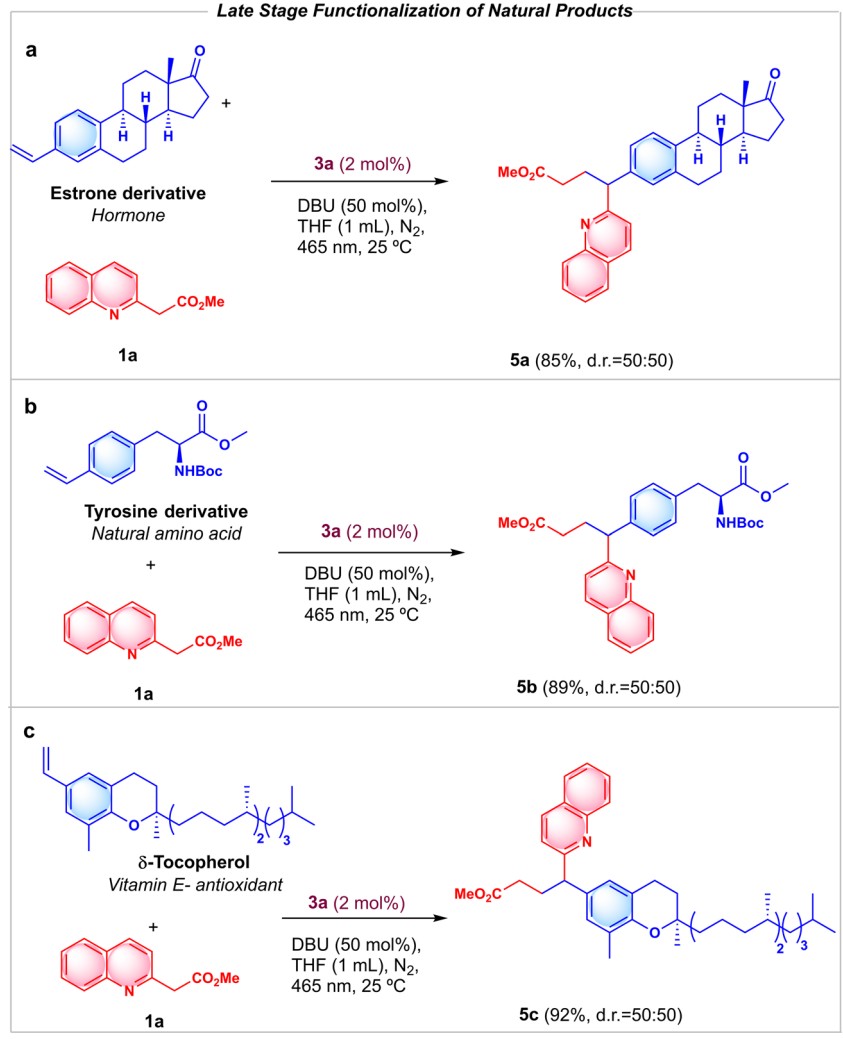

**Fig. 4 Late-stage functionalization using this methodology.** Reaction conditions: 0.1 mmol of **1a**, 0.5 mmol of the corresponding styryl derivative, 0.5 equivalents of DBU and 1.0 mL of THF, under $N_2$, 25 °C, 455 nm LED irradiation (22 W m$^{-2}$) over 17 h. **a** Functionalization of Estrone derivative, **b** Functionalization of Tyrosine derivative, **c** Functionalization of Tocopherol derivative.

slightly underestimate the energy values of $S_0 \rightarrow T_1$ gaps by approximately 10%). Therefore, once the photosensitization of **1a′** takes place, [2 + 2] cycloaddition between $^3$(**1a′**)* and **2a** proceeds. The pathway found consisted of a stepwise addition that produces firstly the 1,4-biradical intermediate **I**, with the unpaired electrons in the most stable positions (benzylic and α to the N). This initial stage of the cycloaddition is both thermodynamically and kinetically very favourable ($\Delta G^{\ddagger} = 7.2$ kcal mol$^{-1}$ and $\Delta G = -17.2$ kcal mol$^{-1}$, Fig. 6). Such a biradical was calculated as both a triplet spin state and an open shell singlet spin state, giving comparable thermodynamic stabilities ($\Delta G = 1.6$ kcal mol$^{-1}$).

Exploration of the further evolution of the open-shell singlet spin state revealed that the intermediate **I** was not very stable because its evolution was even more favourable than its formation. Therefore, the process from **I**, in the open-shell spin state, to the cyclobutane intermediate **II** presents a kinetic barrier of $\Delta G^{\ddagger} = 6.2$ kcal mol$^{-1}$ and its thermodynamics correspond to $\Delta G = -21.9$ kcal mol$^{-1}$. Under the reaction conditions, cyclobutane **II** was not experimentally observed. Instead, a ring-opening rearomatization from **II** accounts for the product **4a** was found. Such a ring-opening process takes place through a kinetic barrier of 17.5 kcal mol$^{-1}$, being the enolic product isoenergetic to intermediate **II**. However, the thermodynamics of this

process is widely compensated by keto-enol tautomerism, with the keto form which is 26.6 kcal mol$^{-1}$ more favourable. Owing to the kinetic barriers calculated, it is inferred that the cyclobutane opening is the actual rate-determining step. In order to corroborate the mechanistic proposal, we performed the reaction with styrene-D8, obtaining **4a-D** with a CD-CD$_2$ fragment (see top-right, Fig. 6).

## Conclusions

In conclusion, we have developed a [2 + 2] cycloaddition and ring-opening rearomatization between electron-deficient methylene azaarenes and double bonds. The reaction gives good results with quinolines, isoquinolines, quinoxalines and pyrazines bearing different electron-withdrawing substituents in the β-position. The process is highly functional group tolerant, and it was performed efficiently with a variety of styrenes, bearing either electron donating or withdrawing groups, as well as heteroaryl or alkyl-substituted double bonds. Moreover, its applicability could be demonstrated by late-stage functionalization of different natural products. Finally, the fluorescence quenching studies together with the theoretical calculations supported a mechanism based on the photosensitization of the *pseudo*-enamine of the azaarene, instead of the alkene derivative.

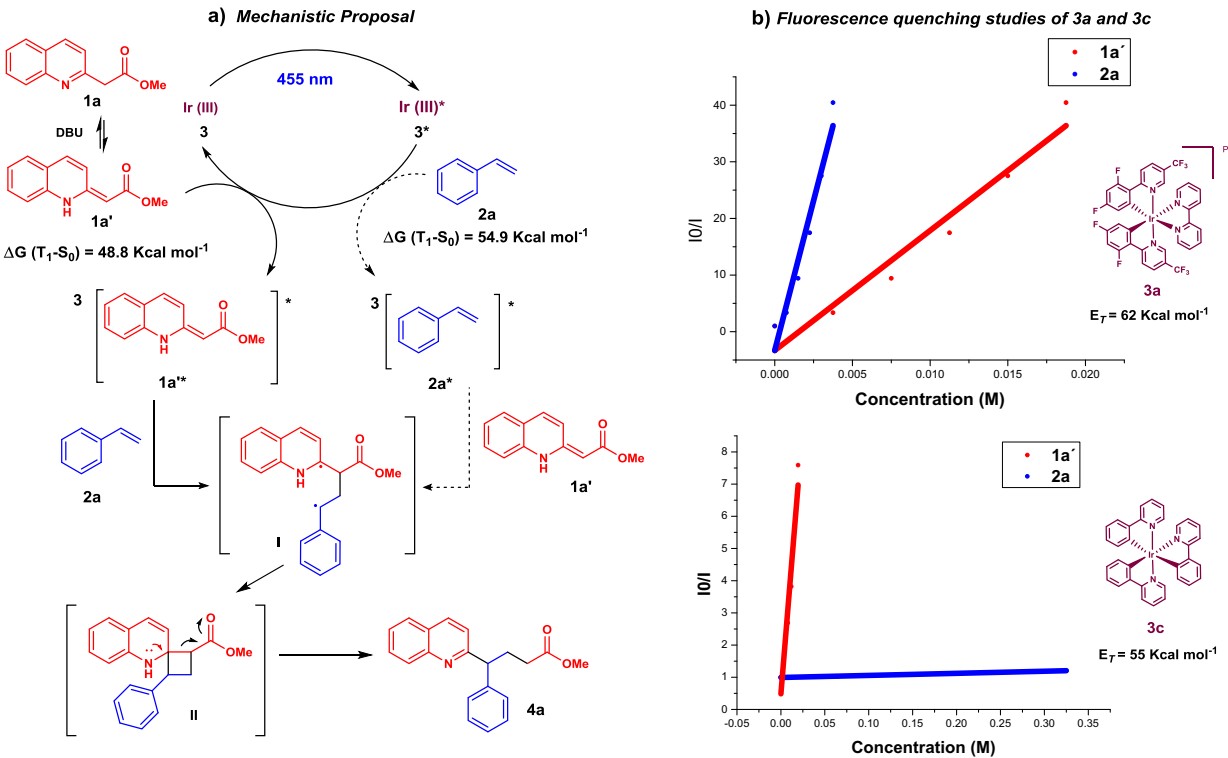

**Fig. 5 Mechanistic studies. a** Mechanistic proposal, **b** Fluorescence quenching studies of **3a** and **3c**.

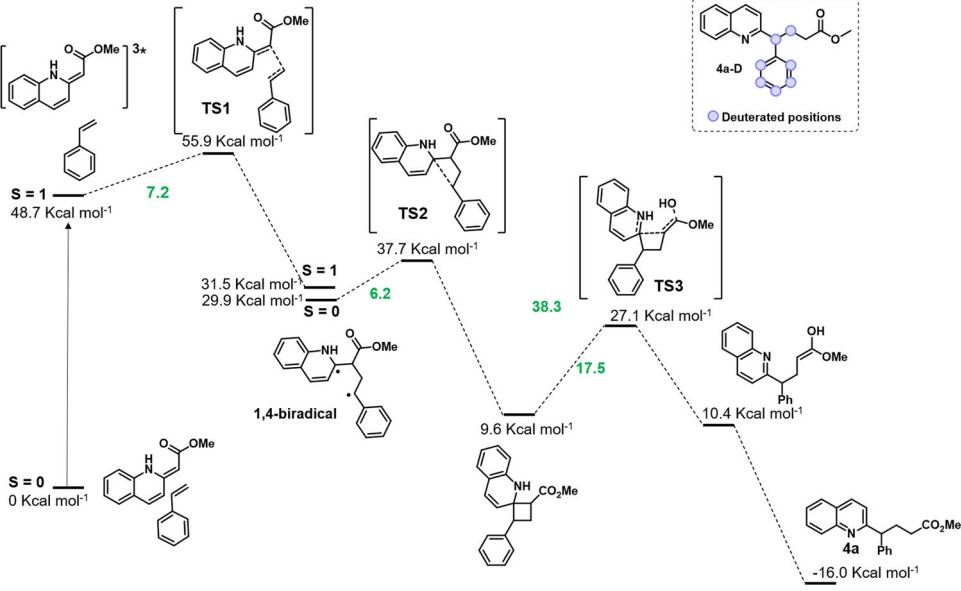

**Fig. 6 Theoretical calculations.** DFT calculations of the reaction mechanism.

## Methods

**General information.** For more details, see Supplementary Methods.

**Detailed optimization of the reaction conditions.** For more details, see Supplementary Note 1 and Supplementary Table 1.

**Synthesis and characterization.** See Supplementary Notes 2–4 and Supplementary Figs 4–99 for NMR spectra.

**Mechanistic investigation.** For more details, see Supplementary Note 5 and Supplementary Data 1.

**General procedures for the photocatalytic reaction**. General procedure A: A dry vial equipped with a magnetic stir bar was charged with **3a** (2.0 mg, 2 µmol, 0.02 equiv.), the corresponding azaarene derivative **1** (0.1 mmol, 1.0 equiv.), DBU (7.6 mg, 0.05 mmol, 0.5 equiv.), alkene **2** (0.5 mmol, 5.0 equiv.) and 1.0 mL of THF (0.1 M). Degasification of the reaction mixture was performed via freeze-pump-thaw cycling (3 × 10 min under vacuum). Then, the reaction mixture was irradiated and stirred in the photoreactor setup at 455 nm for 17 h (unless otherwise stated). The reaction mixture was concentrated under reduced pressure and purified by flash column chromatography (silica gel) to provide the product.

General procedure B: A dry vial equipped with a magnetic stir bar was charged with **3c** (1.4 mg, 2 µmol, 0.02 equiv.), the corresponding azaarene derivative **1** (0.1 mmol, 1.0 equiv.), DBU (7.6 mg, 0.05 mmol, 0.5 equiv.), alkene **2** (0.5 mmol, 5.0 equiv.) and 1.0 mL of THF (0.1 M). Degasification of the reaction mixture was

performed via freeze-pump-thaw cycling (3 × 10 min under vacuum). Then, the reaction mixture was irradiated and stirred in the photoreactor setup at 455 nm for 17 h. The reaction mixture was concentrated under reduced pressure and purified by flash column chromatography (silica gel) to provide the product.

General procedure C: A dry vial equipped with a magnetic stir bar was charged with **3c** (1.4 mg, 2 µmol, 0.02 equiv.), the corresponding azaarene derivative **1** (0.1 mmol, 1.0 equiv.), $K_2CO_3$ (0.1 mmol, 1.0 equiv.), alkene **2** (0.5 mmol, 5.0 equiv.) and 1.0 mL of MeCN (0.1 M). Degasification of the reaction mixture was performed via freeze-pump-thaw cycling (3 × 10 min under vacuum). Then, the reaction mixture was irradiated and stirred in the photoreactor setup at 455 nm for 17 h. The reaction mixture was concentrated under reduced pressure and purified by flash column chromatography (silica gel) to provide the product.

## Data availability
Supplementary Information and relevant data are also available from the authors.

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

## Acknowledgements
We acknowledge the financial support from the Spanish Government (RTI2018–095038-B-I00), CAM_UAM (SI1/PJI/2019-00237), CCC-UAM (computing time), and ERC

(ERC-CG, 647550). L.M. wishes to thank CAM for the 'Atracción de Talento' fellowship. N.S. wishes to thank 'Ministerio de Ciencia e Innovación' for a FPU predoctoral fellowship. The authors also wish to thank the 'Comunidad de Madrid' and European Structural Funds for their financial support to FotoArt-CM project (S2018/NMT-4367).

## Author contributions

N.S. carried out the optimization and scope of the reaction and, in collaboration with L.M., carried out the mechanistic investigation. R.M-B. and L.M. carried out the DFT-computational studies. L.M. and J.A. conceived the project and prepared the manuscript that was edited by all other authors.

## Competing interests

The authors declare no competing interests.
