## [Peer Review File · Communications Chemistry]

Reviewers' comments:

Reviewer #1 (Remarks to the Author):

The manuscript by Marzo, Aleman and coworkers shows an interesting version of the De Mayo reaction for the construction of functionalized and chemically diverse heterocyclic entities. The complementary nature of this reaction with the archetypical Michael reaction for the construction of 1,4-dicarbonyl moieties and the very underdeveloped situation of the De Mayo reaction make these results highly interesting for synthetic chemists. In terms of novelty, one of the authors has already reported the potential of the De Mayo to proceed under photocatalytic conditions (conveniently cited in ref. 25 in the manuscript) but the chemistry shown in this work presents for the first time the possibility of using azomethine-based substrates in the reaction and demonstrate for the first time that the fragmentation of the cyclobutane intermediate through retro-Mannich reaction can operate under photocatalytic conditions. The authors have also demonstrate the wide scope of the reaction with respect to its potential to generate a wide variety of adducts, which also includes its ability to chemically modify several biologically relevant molecules. In addition, mechanistic studies provide a good explanation of the reaction outcome, which is also supported by both experimentation and computational data.

In overall, my opinion is that this manuscript is very well suited for publication in *Comms. Chem.*, as it will gather high attention not only among synthetic organic chemists but also to a wide readership interested in general chemistry. My only recommendation to the authors is to include at least one example showing the performance of the reaction at higher scale, as all examples described in the manuscript show transformations carried out at rather low (0.1 mmol) scale of the limiting reagent.

Reviewer #2 (Remarks to the Author):

Marzo and Alemán present "Visible Light Mediated Photocatalytic [2+2] Cycloaddition/Ring-Opening Rearomatization Cascade of Electron-Deficient Azaarenes and Vinylarenes". In general, the method is simple and can be used for a range of molecules (see substrate scope), including Late Stage functionalization of complex molecules. In addition, experimental section as well as SI were done with care. On the other hand, I feel lack of concept for such a high standard journal requirement. The catalytic system is the same of previous work (see reference 25). It would be nice to see the possibility of an enantioselective version of this process (e.g., *J. Org. Chem.* 2018, 83, 10922.). As a nice extension of Mayo reaction (in fact, racemic Aza-Mayo reaction), I consider this manuscript of great interest of organic chemistry community. Thus, I do recommend the publication elsewhere more specific to this community.

Specific remarks:

1-) Table 1, I believe footnote d) is related to Entry 8.

2-) What is the role for DBU (high impact in the chemical yield, see Table 1)? It should be considered in the Figure 1a and Figure 2 studies as well.

3-) It would be nice if control experiments using D-labelling 1a or/and 2a could be carried out.

4-) Despite the high yield, compounds 5a-c were prepared as diastereomeric mixture 1:1. It seems NMR data in the SI were related to one of those diastereomers.

Reviewer #1

The manuscript by Marzo, Aleman and coworkers shows an interesting version of the De Mayo reaction for the construction of functionalized and chemically diverse heterocyclic entities. The complementary nature of this reaction with the archetypical Michael reaction for the construction of 1,4-dicarbonyl moieties and the very underdeveloped situation of the De Mayo reaction make these results highly interesting for synthetic chemists. In terms of novelty, one of the authors has already reported the potential of the De Mayo to proceed under photocatalytic conditions (conveniently cited in ref. 25 in the manuscript) but the chemistry shown in this work presents for the first time the possibility of using azomethine-based substrates in the reaction and demonstrate for the first time that the fragmentation of the cyclobutane intermediate through retro-Mannich reaction can operate under photocatalytic conditions. The authors have also demonstrate the wide scope of the reaction with respect to its potential to generate a wide variety of adducts, which also includes its ability to chemically modify several biologically relevant molecules. In addition, mechanistic studies provide a good explanation of the reaction outcome, which is also supported by both experimentation and computational data. In overall, my opinion is that this manuscript is very well suited for publication in *Comms. Chem.*, as it will gather high attention not only among synthetic organic chemists but also to a wide readership interested in general chemistry. My only recommendation to the authors is to include at least one example showing the performance of the reaction at higher scale, as all examples described in the manuscript show transformations carried out at rather low (0.1 mmol) scale of the limiting reagent.

Thanks to the referees for the nice words about our manuscript. The output of the photocatalytic systems are very sensitive to the experimental setup. In this work, we have optimized the reaction conditions for a 0.1 mmol scale due our photocatalytic reactor (see supporting information for an image), obtaining excellent results. However, we have tried the reaction in larger scale (1.0 mmol) using different setups but unfortunately the reaction did not work. Different well-known reasons such as the concentration of the reagents and low penetrability of light (Lambert-Beer law) among others, provoke that batch conditions, in general, are not suitable for scaling up photocatalytic reactions (see for a discussion *Chem. Rev.* **2016**, *116*, 10276; *Org. Process Res. Dev.* **2016**, *20*, 4039). Taking into account the importance of developing a scalable procedure, we hypothesized about the use of photocatalytic flow conditions that would be a good solution. However, this will require a deeper study of the reaction conditions and different photo-flow systems. Therefore, we think that this is out of the scope of this work, which is related to show a new de Mayo reaction with Electron-Deficient Azaarenes and Vinylarenes.

Reviewer #2

Marzo and Alemán present “Visible Light Mediated Photocatalytic [2+2] Cycloaddition/Ring-Opening Rearomatization Cascade of Electron-Deficient Azaarenes and Vinylarenes”. In general, the method is simple and can be used for a range of molecules (see substrate scope), including Late Stage functionalization of complex molecules. In addition, experimental section as well as SI were done with care. On the other hand, I feel lack of concept for such a high standard journal requirement. The catalytic system is the same of previous work (see reference 25). It would be nice to see the possibility of an enantioselective version of this process (e.g., *J. Org. Chem.* 2018, 83, 10922.). As a nice extension of Mayo reaction (in fact, racemic Aza-Mayo reaction), I consider this manuscript of great interest of organic chemistry community. Thus, I do recommend the publication elsewhere more specific to this community.

Thanks to the referee for the nice comments about our manuscript. We think that the enantioselective version of this pseudo-enamine reaction de Mayo reaction is a very nice suggestion. However, we think that the standard asymmetric de Mayo reaction has not been developed yet. Both asymmetric reactions would be an excellent matter of study in future research. However, this issue is out of the scope of this manuscript, that has shown, for the first time, the [2+2] Cycloaddition/Ring-Opening Rearomatization Cascade of Electron-Deficient Azaarenes and Vinylarenes.

Specific remarks:

1-) Table 1, I believe footnote d) is related to Entry 8.

Answer: We thank the referee for this comment, and we have now corrected the footnote in table 1.

2-) What is the role for DBU (high impact in the chemical yield, see Table 1)? It should be considered in the Figure 1a and Figure 2 studies as well.

Answer: The DBU is shifting the equilibria to the formation of the pseudo-enamine **1a'**, thus promoting the [2+2] cycloaddition reaction to take place. We have now included its role in Figure 1.

3-) It would be nice if control experiments using D-labelling 1a or/and 2a could be carried out.

Answer: We have carried out the reaction with styrene-D8 and we have observed the formation of the deuterated product (4a-D, see below), supporting our proposed mechanism. This information have been included in the main text of the manuscript and the compound has been characterized (see supporting information). In addition, the formation of this specific regioisomer was determined by bidimensional NMR experiments that have been now included in the SI.

● Deuterated positions

4-) Despite the high yield, compounds 5a-c were prepared as diastereomeric mixture 1:1. It seems NMR data in the SI were related to one of those diastereomers.

Answer: As the referee points out the NMR spectrum suggest the formation of a single diastereoisomer, but SFC analysis of the compound **5b** shows the formation of two diastereoisomers in 1:1 relationship (included in the SI, see below). The *dr* for compounds **5a** and **5c** were assumed by analogy. In addition, for these type of molecules a very similar behavior has been previously reported in the literature (see e.g. M. Mato, B. Herlé, A. M. Echavarren, *Org. Lett.* **20**, 4341–4345 (2018)).

REVIEWERS' COMMENTS:

Reviewer #2 (Remarks to the Author):

Accept.